# Economic Problems and Loneliness as Factors Related to Subjective Unmet Health Needs in People with Chronic Diseases and Dependency

**DOI:** 10.3390/ijerph17082924

**Published:** 2020-04-23

**Authors:** Olga María López-Entrambasaguas, José Manuel Martínez-Linares, Miguel Sola-García, Carmen García-Redecillas, Ana María Díaz-Meco-Niño

**Affiliations:** 1Department of Nursing, Universidad de Jaén, 23071 Jaén, Spain; omlopez@ujaen.es; 2Complejo Hospitalario de Jaén, Servicio Andaluz de Salud, 23007 Jaén, Spain; misola1989@gmail.com (M.S.-G.); garciaredecillas@gmail.com (C.G.-R.); adimeni64@hotmail.com (A.M.D.-M.-N.)

**Keywords:** needs assessment, disabled persons, qualitative research, chronic disease, home nursing, vulnerable population

## Abstract

*Background:* The continuous increase of people with chronic diseases is one of the greatest challenges for healthcare systems worldwide. Population growth and life expectancy means that an increasing number of people with chronic diseases and dependency need some kind of assistance to meet their needs. Determining these subjective unmet needs helps to understand the situation of these people. The aim of this study was to explore the perceptions of chronic patients over 65 years of age from the day-care center toward subjective health needs that are not being met by the socio-health system. *Methods:* Qualitative exploratory-descriptive study. Through convenience sampling, we selected people with chronic diseases and dependency who used day-care centers and met the inclusion criteria. Focus groups were performed. The data were transcribed and a thematic analysis was carried out using Atlas.ti software. *Results:* The topics resulting from the analysis were classified into dissatisfaction of biological/physiological needs, psychological needs, social needs, and other issues that arose in both groups of participants which referred to the types of needs previously indicated. The issues related to social and psycho-social needs stood out. *Conclusions:* People with chronic diseases and dependency have their physiological needs covered with the help they receive, but their situation of dependency generates additional costs that worsen their economic situation. However, their greatest need is due to the loneliness they feel and the feeling they have of “being a burden” on their families.

## 1. Introduction

The healthcare systems of developed countries are facing the challenge of providing attention to an increasing number of people with chronic diseases, due the increase of life expectancy and the aging of the population caused by the decrease in birth rates. A chronic disease is any long-term health issue and, generally, of slow progression [1]. According to a study at the European scale, Switzerland is the country with the best data about chronic diseases and Belgium is the opposite end of the rank [2]. These data can be extrapolated at the international level, which is why the World Health Organization has proposed the goal of reducing the premature mortality by chronic diseases by 25% in the year 2025, under the “25 × 25” motto [3].

The data provided by the European Health Interview Survey of 2014 [4] indicate that 59.83% of the Spanish population over 15 years of age suffer from a chronic disease, with the most frequent diseases being high blood pressure (18.74% of cases), hypercholesterolemia (16.70%), chronic cervical pain (16.52%), chronic low back pain (19.22%), diabetes mellitus (6.84%), and chronic respiratory diseases (3.32%).

The condition of being chronically ill produces common consequences among those who suffer from it: difficulty in decision-making, mood alteration, communication problems in the doctor–patient relationship and in the family, labor and social scope, lack of adherence to treatments or medical guidelines, feeding problems, alteration of the physical functions, lack of physical activity, fatigue, and difficulties with pain management, among others [5,6].

Both the increase of the rate of chronic diseases and their causes have raised the number of people in a situation of dependency. That is, there is a larger number of elderly people with greater risk of suffering from chronic diseases, which can trigger complications that such people survive, although leaving them in a situation of dependency [7]. This creates a relationship between chronic diseases and dependency, giving rise to the existence of people with chronic diseases and dependency (PCDD).

In the year 2019, the rate of dependency in the Spanish population was 54.28%, which had increased continuously since the year 2008 [8]. In the case of people over 64 years of age, this rate was 29.94%, also with a positive increase since 2008 [9]. Day-care centres are part of the assistance offer for people with dependency. There are 3387 day-care centers in Spain, with the capacity to attend to a total of 90,577 people. Sixty percent of the assistance is financed with public funds, and 72,897 of the people attended in these centers are PCDD [10].

From the healthcare perspective, the most important subjective unmet needs are those perceived by people with chronic diseases, since they can cause delays in the reception of healthcare services and, in turn, worse health results [11]. Therefore, determining the relationship between the unmet healthcare needs and the adverse results is important from the point of view of healthcare service provision, since the detection and removal of potentially modifiable barriers to attention may improve the health results. However, there is limited and inconsistent evidence on the relationship between the needs that are not satisfied by the healthcare system and the health results [12].

Some previous works have associated subjective unmet needs with an increase in the visits to emergency services [13,14], whereas other studies show inconclusive data of the rates of hospital admission and general visits to the doctor within the general population [15]. Few studies have addressed the effect of unmet needs and their consequences in a high-risk population of patients with chronic diseases. A study on such needs is fundamental to perform a diagnosis of the situation, as well as to establish what services and attention are being provided, which of these are adequate, and whether there are deficiencies, regarding PCDD.

A study on subjective unmet needs is fundamental to know the services that are being provided, the ones that are most adequate and their deficiencies, from the perspective of PCDD. Therefore, unravelling the subjective unmet needs of these people will allow developing programs and strategies designed to that end and, thus, contributing to improving the results in health and quality of life and to promoting a better use of social and healthcare services.

The aim is to explore the perceptions of PCDD over 65 years of age and who are users of day-care centers toward subjective healthcare needs that are not being satisfied by the social health system.

## 2. Materials and Methods

The methodology used in this study is presented in the Consolidated Criteria for Reporting Qualitative Research (COREQ) format [16] for qualitative studies.

### 2.1. Research Team and Flexibility

The members of the research team had the following training: J.M.M.L. (male), nurse and doctor; O.M.L.E (female), nurse and doctor; M.S.G (male), family physician and doctor; C.G.R. (female), geriatric specialist physician; A.M.D.N. (female), nurse and doctorate student. During the study period, J.M.M.L. and O.M.L.E. worked as faculty members, M.S.G. and C.G.R. as a family physician and a geriatric specialist physician, respectively, and A.M.D.N. as a nurse. All the members of the research team had research training and they all had previously participated in both quantitative and qualitative research projects. None of the research team members had any kind of relationship or previous contact with the participants of the study. The latter were informed about the composition of the research team and about the aim and interest of the study when they were offered to participate in it.

### 2.2. Study Design

A descriptive-exploratory qualitative study was carried out. This is the most suitable methodology due to the complexity of the aim of the study and the difficulty in measuring the concepts [17], and it allowed obtaining the subjective perception of the participants [18]. This methodological approach is based on the principles of naturalistic observation, which aims to study individuals in their natural state [19]. Through convenience sampling, we recruited PCDD who met the following inclusion criteria: to have a moderate (40–55 points) or severe (20–35 points) degree of dependency according to the Barthel Index [20], translated into Spanish [21]; to be a user of a day-care center, where the contact with and recruitment of the participants took place, and to have no cognitive deterioration. The Barthel Index is a validated instrument that allows assessing the level of dependency of a person to carry out basic activities of daily living, assigning scores based on his/her capability.

The sample was composed of a total of 23 PCDD from the localities of Martos and Mancha Real (Jaén, Spain) (17 people of Martos and 6 people of Mancha Real). The data gathering was conducted in a designated room in the day-care centers of these localities. No additional people other than the participants and the members of the research team were present during the data-gathering process.

The main characteristics of the participants are detailed in Table 1.

Three focus groups were performed (two in Martos and one in Mancha Real) in April and May 2019, with which data saturation was reached [23]. The ad hoc script of questions was designed to obtain the information required to respond to the proposed objective, and it was created and revised by all the members of the research team. A pilot focus group was conducted with three PCDD in March 2019. Then, some questions were re-written, added, or removed, resulting in the final version of the script. Table 2 shows the main questions included in the focus groups and in the interviews.

None of the focus groups had to be repeated. All of them were recorded in audio, and field notes were taken, which were also incorporated in the data analysis. The duration of the focus groups was in the range of 1.5–2.5 h (including some breaks). The transcriptions were not given to the participants, since they did not want to revise them.

### 2.3. Data Analysis and Results

The coding of the transcriptions was conducted individually by two researchers, who then unified their encoded transcriptions to produce the final coding [24]. The content of the transcriptions was analyzed following the method of the six phases of thematic analysis with scientific rigor described by Braun and Clarke (familiarization with the data, generation of initial categories or codes, theme search, theme revision, definition and naming of themes, and writing of the final report) [25], as well as the process to ensure the reliability of the results described by Nowell et al. [26]. The codes that were related to each other were grouped in categories, from which subthemes emerged, which in turn were grouped to produce the final themes. This qualitative analysis of data was conducted using Atlas.ti v.5 for Windows^©^ (ATLAS.ti Scientific Software Development GmbH, Berlin, Germany). The results were not sent to the participants for revision.

### 2.4. Ethical Considerations

The study was carried out following the ethical principles of the Declaration of Helsinki. The treatment of personal data was performed in compliance with Regulation 2016/679 of the European Parliament and the Council of April 27th 2016, on the protection of natural persons with respect to the treatment of personal data and the unrestricted movement of such data, which revokes Directive 95/46/CE.

This study was conducted after obtaining the approval from the Ethics Committee of the University of Jaén. Each participant was requested to sign the corresponding informed consent.

## 3. Results

The generated themes were classified into the following categories: biological needs, psychological needs, social needs, and socioemotional needs.

### 3.1. Biological Needs Results

The results related to the biological needs of PCDD did not include the existence of unmet physiological needs, such as eating, going to the bathroom, moving, or sleeping, among others; these results referred to the needs of assistance material to satisfy this type of needs (Table 3).

Therefore, a theme emerged about the needs of assistance material to carry out basic daily life activities or to satisfy physiological needs. The PCDD had assistance material provided by the public healthcare system, such as walking frames, wheelchairs, and crutches.
“Is that walking frame yours? No, the government gave it to me when I got that thing in my leg and I could no longer walk on my own”(GF2PC-P2)
“And do you have a wheelchair at home, or a walking frame …? I have been in this wheelchair for four years, which was given to me by the Social Security”(GF1PC-P5)

However, some material is not provided by the public healthcare system in cases of extreme need, such as articulated beds, and an extra price must be paid to obtain material of better quality than that provided by the government.
“My daughter, who is the one who takes care of me, finds it hard to move me. She talked to the people at the Social Security to see if they can give us a bed … They told her that that is for people who are permanently in the bed. But it is very hard for her to move me! And I suffer, because I see that she struggles …”(GF1PC-P6)
“I did get a wheelchair, and the walking frame too, but I paid a little extra and I got this one, which is better”(GF3PC-P4)

### 3.2. Results Related to Psychological Needs

The participants perceived that, emotionally, they had experienced a series of changes that generated a series of needs, since they felt that they had lost the authority they once had, they refuse to leave their homes and, above all, they feel lonely and sad (Table 4).

The participants perceived that they had lost the authority that they used to have in the past, since they think that the socio-sanitary system does not attend to their demands and does not consider their opinions. Other people make the decisions for them, and they can even find this offensive.
“Offended, I feel offended. They don’t listen to me!”(GF1PC-P1)
“The cardiologist tells me: Make sure they don’t give you a different medicine; they must give you the same one! Then why do they change it in the pharmacy after I already told them?”(GF3PC-P2)

These people cling on to the idea of staying in their homes. Those who had moved out highlighted that they missed living in their own place.
“I loved being in my house… (now she lives with her daughter)”(GF1PC-P3)
“My daughter tells me that I should move in with them. But they are working! They leave at 7 in the morning and come back at 3 in the afternoon! What do I do in their house alone? I’m better off in my house”(GF1PC-P4)

In this regard, there was a generalized feeling of being “at ease” during the time they spend in the day-care center, although they were also concerned about being a burden on their children, which leads them to prefer being alone in their own homes.
“They treat me very well in this centre … That’s all I can say”(GF2PC-P3)
“We need more attention from our relatives, but then they can’t have a normal life. They help me with everything I do, but I don’t want to be a burden on them, because I suffer when I see that they can’t do what they want to do or go wherever they would like to go because of me”(GF1PC-P1)

Lastly, the participants described a series of shared feelings, which are not attended to from the public socio-sanitary system. These are related to sadness, nostalgia, and especially loneliness.
“It’s not the same anymore. When my husband was alive, we used to go traveling, and sometimes our children came with us …”(GF3PC-P4)
“I think I need somebody who could be with me at times, to go for walks … I feel very lonely!”(GF2PC-P4)

### 3.3. Results Related to Social Needs

Most of the needs mentioned in the analysis of the results were related to the social sphere of the PCDD. All of them were grouped into two themes: economic unmet needs and critiques on the socio-sanitary system (Table 5 and Table 6).

With respect to finances, the participants usually lose control on their personal and home economy, and it is their children who take such control, as well as control of the rest of the bureaucratic processes that need to be done. This was highlighted by the PCDD in the questions related to the expenses that could derive from their situation.
“My children manage everything related to money. I ask them and they tell me that everything is paid through the bank. And the water and electricity bills and all that... my children arranged everything to be paid through the bank, and they manage my accounts”(GF2PC-P5)

The participants perceived that they had one or more chronic diseases that generate a moderate or severe degree of dependency, with the subsequent additional expenses. Some of these expenses are the copayment of the day-care center and the need to hire an informal caregiver in addition to the one who, sometimes, is provided by the public administration. This assistance does not cover their daily needs and they must hire such service privately, usually through informal caregivers with no training. If their pensions would increase, they stated that such increase would be assigned to help their children economically in order for them to have more time to be with them, and to hire informal caregivers to meet their needs.
*“I have to pay a woman to take care of me, and I also have to pay the day-care centre**—**How much do I have to pay? I don’t know. I pay through the bank, but I know that I have to pay for things that I wouldn’t have to if I wasn’t in this state”*(GF2PC-P6)

Therefore, the PCDD perceived that their diseases affected them economically and that there is a lack of public aid to improve their situation.
“Do you think a couple can live on a pension of 800€? We have to pay electricity, water, this day-care centre … We don’t have enough resources!”(GF3PC-P1)

There were a series of critiques on the current socio-sanitary system related to three specific scopes: social benefit system, healthcare system, and caregivers. Firstly, in their opinion, the current system of social benefits does not provide a solution to their problem and some people do not get the help they need; some people wait a long time to receive such help.
“We have been waiting for four years to get this help … but it just doesn’t arrive”(GF2PC-P6)
“I only want one of those beds. It would be of great help for my daughter when she has to move me. That’s all I’m asking for”(GF1PC-P6)

Secondly, the participants expressed their critiques, and compliments, on the attention provided to them by the public healthcare system when they make use of it. Their complaints were focused on the lack of material and professional resources with which this system attends to them. However, they highlighted the good treatment they received from the health personnel who attended to them. To this respect, the management of medical visits and revisions is assumed by their children or carers.
“No, it’s not enough with the people available to provide attention. They treat us well, but there should be more professionals in the day-care centre when we go there”(GF3PC-P1)
“My children get the prescriptions for me … for the diapers and medicines I need and all that”(GF1PC-P4)

Lastly, the participants also expressed critiques on the attention they receive from the caregivers provided by social services, since some PCDD felt disgraced for not having such support personnel. However, they also believed that these professionals are well-trained to carry out their job, which they do the best way they can.
“That person comes three days per week. She helps me with whatever I can’t do on my own. She helps me with my medicines and my shopping bags … she helps me with the house … because I can’t do it myself anymore”(GF2PC-P4)

### 3.4. Results Related to Psychosocial Needs

The participants also talked about their psychosocial needs and the relationship between them and their families. The two subthemes generated from this theme were the relief they get from the fact that their relatives take care of their bureaucratic processes and their perception of being a “burden” on their children, which appears again, tackling their family relationships (Table 7).

The processes and paperwork that PCDD need to perform are managed and solved by their relatives, who, sometimes, must make decisions for them, which increases the perception of authority loss, as was previously mentioned.

Furthermore, as was stated in their unmet emotional needs, PCDD had the feeling of being a “burden” on their children. The latter are committed to their care, but their children have other obligations that prevent them from providing them all the attention they need. That is what leads PCDD to hire private informal caregivers or the services of a day-care center. All this causes the situation of chronic disease and dependency to have an impact on the family and economy of the people who suffer from it.

## 4. Discussion

The World Health Organization defines health as “a state of complete physical, mental and social well-being, and not only the absence of affectations or diseases” [27]. However, the models from which health is approached have changed, that is, from the Biomedic Model, which reduces disease to the deviation of a series of biological (physiological) variables, to the Biopsychosocial Model proposed by Engel [28], which is framed within the General Systems Theory and for which there are multiple causes of health that comprise the biological, psychological, and social spheres of the individual. From this basis, the present study was focused on exploring the subjective unmet healthcare needs of PCDD, addressing those three spheres that influence the individual and his/her health.

The care that PCDD require to satisfy their healthcare needs has changed in the last years with regard to the person who provides it. In the last 20 years, the number of potential carers per PCDD has decreased from 15 in the year 1998 to 9 in the year 2018. In the case of carers aged 40–64 years, 85% are women, although this gender breach inverts with older carer age, which is up to 75% males in the group of carers of 90 years of age and older [29].

The different European countries have systems of attention to dependence based on four models: liberal model (British Islands), Nordic model, corporate model (Central European countries), and Mediterranean model. However, they all share a series of characteristics, such as the fact that they do not replace the work performed by the families of PCDD, they provide economic aid, residential attention, and home services, they are publicly regulated and funded, and the users participate in the financing of these services [30].

The participants did not mention the existence of unmet biological or physiological needs, such as eating, washing up, moving, etc. They had the help of formal and/or informal caregivers and relatives to cover those needs. In this scope, the demand was related to the acquisition of complex material of technical aid (mainly adjustable beds and cranes) and assistance to make the necessary adjustments in the home of the PCDD in order to be able to use other technical aid that require more space [31]. Simpler technical aid is easily and quickly provided (walking frames, crutches, wheelchairs, etc.), but more complex technical aid and home adjustments are more slowly provided. All this technical assistance would help the person or his/her carer to satisfy this type of needs.

In the psychological sphere, the participants detected that their situation made them lose the authority they had in the past. This perception was shared and is reported in other studies, even in people who are not in a situation of dependence. An example of this is the qualitative study conducted with people in a situation of pre-retirement aged between 50 and 65 years, who described a loss of authority to their children, since their provider role deteriorated and they had to rely on other factors such as complicity, sharing daily moments more often and participating in the home tasks more frequently [32]. The retirement age seems to be the moment when the loss of authority of a person begins, since the fact of being no longer productive and useful is not limited to the social scope, as it also involves the family sphere.

The authority of age is losing relevance due to the fact that the importance of elder people as passers of culture for the younger generations is decreasing; the oral information that these people used to transmit has been replaced with mass communication through the use of new technologies, and elder people have been isolated, with many of them being moved to hospices [33]. Culturally, society has shifted from a model based on family values and the recognition of the authority of elder people to a more recent model based on economic growth, constant change, and consumption [34].

However, the most relevant subtheme derived from the analysis, with a generalized consensus, was the loneliness that these people feel, which has become an endemic harm of the society of developed countries. In Spain, 43.1% of the homes are inhabited by a person of 65 years of age or older who lives alone; that is, 2 million people are in such situation [35]. This means 3.9% more with respect to the previous year. Although there are elder people who prefer to live alone, which can be an indicator of success, independence, and well-being, many others live this loneliness with anxiety.

Unwanted loneliness affects the health and quality of life of elder people. People who live alone unwillingly can have a higher risk of premature death, by up to 14% [36], since feeling alone is worse than being alone [37], and loneliness is an important factor to understand the development of mental health problems in this population [38].

On the other hand, other elders decide to live alone to avoid the feeling of being a “burden” to their children [39], although there are more causes that contribute to such perception, such as the increase of expenses derived from a certain degree of dependence [40], the perception of the deterioration of their quality of life and the loss of autonomy [41], and the belief that they interfere with the work and social life of their relatives [42]. These results are in line with those obtained in the present study.

It is a fact that people in a situation of dependency have additional expenses, which increase with the degree of the dependency, due to the technical aid required, the adjustments that need to be done in their homes, the hiring of caregivers, etc. The budget cuts on the Dependency Law [22] left those needs unmet. Therefore, according to the studies conducted to this respect, the total estimated cost of 4.193 billion hours of care provided by 1,326,270 informal carers in Spanish homes in 2008 would have posed an expenditure of 23.064 to 50.158 billion euros from the public administration, that is, between 2.1% and 4.6% of the gross domestic product of that year [43]. This fluctuation is due to the method used to estimate such cost.

The PCDD and their families are forced to pay these expenses, with the subsequent repercussions on the health state (physical, psychological, and social) of the informal carers [44,45,46]. A comparative study carried out in the UK, United States, and Spain showed that the percentage of people with functional limitation who receive informal care is higher in Spain than in the other two countries. Moreover, such care is provided by people from outside of the family circle at a higher percentage in United States and the UK with respect to Spain [47]. All these studies, which are in line with the results obtained in the present study, also demonstrate that the family circle and the involvement of the family members in the provision of care constitute a fundamental support for PCDD.

Thus, the present study shows the importance of informal caregivers in the ethical, social, and economic scopes. Therefore, the public administrations and society in general must pay more attention to them and give them greater recognition.

The importance of the Dependency Law [22] became evident with the arrival of support for PCDD, who saw the satisfaction of previously unmet needs. However, due to the unequal implementation in the whole of Spain and the lack of budget allocation, the demand generated was not completely satisfied. The study of the Spanish Federation of Municipalities and Provinces [48] highlights that this has led to a constant increase in the waiting lists. This law was passed by the Spanish Parliament to provide a response to people who, due to their situation of dependency, require assistance to carry out the basic activities of daily living. This law regulates the attention to people in a situation of dependency through the creation of a system for autonomy and attention to dependency.

A country is considered to have an aged structure when the proportion of people of 60 years of age or older reaches 7%. In the year 2019, such percentage in Spain was 19.40%, after a continuous increase from the year 2009, being the highest since the first records [9]. In addition to this, the dependency rate of the Spanish population over 64 years of age is 29.94% [49]. Some studies even predict that, in the year 2050, there will be in Europe more PCDD than people who can provide the care and support they need [50].

In view of these data and the results of the present study, actions and strategies should be designed and implemented to contribute to improving the assistance given to PCDD in their homes. The conditions of suffering from a chronic disease and being in a situation of dependency generate a series of needs that must be satisfied with the help of other people. To respond to this, the healthcare systems and social services need to ensure their capability to develop and coordinate multidimensional care models that include the necessary professionals who can meet the needs of PCDD, especially taking into account the demographic challenge and the aging of the population that society is facing. We need a model of attention focused on the person and based on the principles of the humanistic theory, in which the person is the center of any intervention, according to Maslow’s hierarchy of human needs.

Although the definition of health provided by The World Health Organization has not been modified since the year 1948, the needs of the population have changed, as well as the expectations they generate regarding the healthcare systems and social services with respect to the support they will have to be able to provide. Therefore, further studies should be conducted in this research line to demonstrate these aspects.

This study shows that PCDD do not perceive that they have physiological unmet needs. They focus their requests on technical home assistance materials (adjustable beds and cranes, mostly) that facilitate the job of the people who provide the assistance service. Emotionally, their needs are related to their feeling of loneliness and their self-perception of being a “burden” to their relatives. Satisfying these needs implies an additional expense, which increases proportionally with the degree of dependency. All these perceptions are a cause of criticism against the current socio-sanitary system.

### Limitations

This study shows the characteristic limitations of a qualitative study, thereby the data generated here cannot be extrapolated to other social and economic contexts due to the differences in the results that could derive from such extrapolation. Similarly, different results could be obtained if such study was carried out in the same environment but in a different time.

The fact that we recruited PCDD in day-care centers implies that the results could be different if the participants were permanently in their homes or coexisted with their children. Likewise, the results could be different if the participants´ degree of dependency were mild.

## 5. Conclusions

The participants did not have unmet basic physiological needs. Their unmet needs were related to complex technical assistance material. In the emotional scope, their unmet psychological needs were generated by the loss of authority, clinging on to their homes, the feeling of loneliness, and the feeling of being a “burden” on their children. Socially, the needs they described were related to economic problems caused by the additional expenses derived from their situation of dependency and the lack or delay in the provision of public support.

## Figures and Tables

**Table 1 ijerph-17-02924-t001:** Sociodemographic characteristics of the sample of people with chronic diseases and dependency (PCDD).

Code	Sex	Age (Years)	Chronic Pathologies	Evolution Years	Degree of Dependency (Barthel Index Score)	Recognition of Situation of Dependency *	Assigned Carer	Civil State	Number of Children	Lives Alone or with One or More People
GF1PC-P1	Male	89	Parkinson’s disease and polyarthrosis	37	Moderate (55)	Yes	Yes	Widow	3	Alone
GF1PC-P2	Female	84	Polyarthrosis	8	Moderate (55)	Yes	Yes	Widow	2	Alone
GF1PC-P3	Female	84	Stroke and cardiopathy	7	Moderate (45)	No	Yes	Widow	4	Alone
GF1PC-P4	Female	86	Polyarthrosis	32	Moderate (55)	Yes	Yes	Widow	3	Alone
GF1PC-P5	Female	91	Parkinson’s disease and polyarthrosis	10	Moderate (55)	Yes	Yes	Widow	1	Alone
GF1PC-P6	Female	82	Miastenia gravis and DR	12	Moderate (55)	Yes	Yes	Widow	3	Alone
GF1PC-P7	Male	89	Polyarthrosis and UI	11	Moderate (55)	No	No	Widow	1	Alone
GF1PC-P8	Female	76	Stroke	13	Severe (35)	Yes	Yes	Widow	4	Alone
GF2PC-P1	Female	85	Polyarthrosis	10	Severe (20)	Yes	Yes	Married	3	Not alone
GF2PC-P2	Female	81	Polyarthrosis and UI	10	Moderate (55)	No	No	Widow	4	Alone
GF2PC-P3	Female	84	Stroke	12	Severe (25)	Yes	Yes	Widow	3	Alone
GF2PC-P4	Female	87	Stroke	8	Moderate (55)	No	No	Widow	4	Alone
GF2PC-P5	Female	80	COPD and cardiopathy	7	Moderate (55)	No	No	Widow	4	Not alone
GF2PC-P6	Male	91	Polyarthrosis and cardiopathy	11	Moderate (55)	No	Yes	Widow	5	Alone
GF2PC-P7	Female	91	Polyarthrosis and cardiopathy	8	Moderate (55)	No	Yes	Widow	2	Alone
GF3PC-P1	Female	83	Polyarthrosis and cardiopathy	7	Moderate (55)	Yes	Yes	Married	4	Not alone
GF3PC-P2	Female	84	Stroke	6	Moderate (55)	Yes	Yes	Widow	1	Not alone
GF3PC-P3	Female	78	COPD and cardiopathy	10	Moderate (55)	No	Yes	Married	2	Not alone
GF3PC-P4	Female	84	Polyarthrosis	12	Moderate (55)	No	Yes	Married	3	Not alone
GF3PC-P5	Female	82	COPD and DR	30	Moderate (55)	Yes	Yes	Married	5	Not alone
GF3PC-P6	Female	85	COPD and polyarthrosis	12	Moderate (50)	Yes	No	Widow	2	Alone
GF3PC-P7	Male	84	Stroke	9	Moderate (50)	No	No	Married	2	Not alone
GF3PC-P8	Female	90	Polyarthrosis and cardiopathy	15	Moderate (55)	No	Yes	Widow	2	Alone

Abbreviations: COPD, chronic obstructive pulmonary disease; UI, urinary incontinence; PCDD, people with chronic diseases and dependency; DR, diabetic retinopathy. * Recognition of the situation of dependency according to Law 39/2006, of December 14th, on the Promotion of Personal Autonomy and Attention to People in a Situation of Dependency [22]. Source: developed by author.

**Table 2 ijerph-17-02924-t002:** Main questions asked in the focus groups and interviews with PCDD.

Pre-Established Categories	Questions for PCDD
Basic needs	Do you think your basic needs are met (eating, dressing, moving, going to the toilet…)? What help do you require for these to be satisfied?
Attention provided by the carer	What are the tasks of the carer in your home? Do you think that the time the carer spends in your home is enough to meet your needs? Do you think that the carer is sufficiently prepared to attend to both your daily needs and emergency situations that may occur in your home?
Availability of assistance material in the patient’s home	Do you have any sort of sanitary and/or assistance material in your home due to your chronic disease? If so, has it been provided by the Social Security or social services, or did you have to buy it? Do you think that the material you have is enough or do you require further material? If so, which material?
Bureaucratic management and processes	Do you need somebody to perform these processes? Who is that person: relative, carer, neighbor, friend? Why do you need this kind of help? What perception do you have toward the difficulty of carrying out these processes?
Personal and family economy	Would you rate your current economic situation as worse, same or better with respect to when your chronic disease began? If there is a difference, what do you think this difference may be due to? Do you think that your chronic disease has deteriorated your economic situation?
Emotional attention	How do you think your chronic disease has emotionally and socially influenced your daily life? Have you ever needed assistance/therapy in the psychological/emotional aspect due to your chronic disease?
Contribution of solutions	To respond to unmet needs, what solutions do you think that could be carried out?
Other needs	Apart from the personal, material and emotional/social needs, what other needs do you have as chronic patients that have not been mentioned yet?

Abbreviations: PCDD, people with chronic diseases and dependence. Source: developed by author.

**Table 3 ijerph-17-02924-t003:** Theme, subthemes, and codes related to the satisfaction of biological needs of PCDD.

Theme	Subtheme	Codes
Needs of assistance material	Assistance material provided by public services	Assistance material provided
		Walking frames provided
		Wheelchairs provided
		Other assistance material provided
		Contribution for better quality material
	Unmet needs related to assistance material	Assistance material not provided
		Assistance material to purchase
		Adjustable beds not provided
		Cranes not provided
		Other assistance material not provided

Source: developed by author.

**Table 4 ijerph-17-02924-t004:** Theme, subthemes, and codes related to the satisfaction of psychological needs of PCDD.

Theme	Subtheme	Codes
Emotional needs	Loss of authority	Feels unheeded
		No solutions are offered to solve his/her problems
		His/her opinion does not count
		His/her relatives decide for him/her
	Clinging on to their homes	Does not want to leave his/her home
		Misses his/her home
		Moving to a different home
		Periods in the homes of his/her different children
	Loneliness and sadness	Feeling of sadness
		Feeling of loneliness
		Nostalgia for times past
		Nobody visits him/her

Source: developed by author.

**Table 5 ijerph-17-02924-t005:** Theme, subthemes, and codes related to the satisfaction of social needs of PCDD.

Theme	Subtheme	Codes
Economic unmet needs	Loss of control on their own economy	Does not know the expenses
		His/her relatives manage his/her economy
		Does not control his/her expenses
	The situation of dependency generates additional expenses	Day-care center expense
		Expense for hiring private home assistance
		Pension is not raised with dependency
		Paying for home assistance
		Cannot help his/her children financially
	Lack of public aids	Complains about the lack of public aid
		Would spend more in private assistance if he/she could afford it
		Children cannot help financially

Source: developed by author.

**Table 6 ijerph-17-02924-t006:** Theme, subthemes, and codes related to the satisfaction of social needs of PCDD (continuation).

Theme	Subtheme	Codes
Critiques on the socio-sanitary system	Critiques on the social benefit system	Benefits for the basics
		Delayed reception of benefits
		No effective solutions are offered
		People who do not receive home assistance
	Critiques (and compliments) on the healthcare system	Good treatment but not enough staff
		Complaints about the assistance received
		Delayed visits from the specialists
	Critiques (and compliments) on the caregivers	Carer with work overload
		Carer who is well-trained to do their job
		People who do not have a home carer
		Carer who cannot go beyond his/her capacity
		Carer who knows how to do his/her job

Source: developed by author.

**Table 7 ijerph-17-02924-t007:** Theme, subthemes, and codes related to the satisfaction of psychosocial needs of PCDD.

Theme	Subtheme	Codes
Family relationship	Solved bureaucratic processes and management	His/her relatives do the paperwork
		His/her relatives solve his/her bureaucratic problems
		Unconcern about paperwork
		Does not know how to solve bureaucratic problems
	Feeling of being a “burden”	Relatives with little time to attend to them
		His/her situation affects his/her family members
		Does not want to be a nuisance
		Perceives him/herself as a burden
		Would like to need no help

Source: developed by author.

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
