# Peer review of "Economic Problems and Loneliness as Factors Related to Subjective Unmet Health Needs in People with Chronic Diseases and Dependency"

_ijerph, 2020, doi:10.3390/ijerph17082924_

Round 1
Reviewer 1 Report
ummary: this study aims to explore subjective unmet health needs through conducting a focus group survey for 23 elderly chorionic patients in day – care center. Qualitative exploratory – descriptive study is analyzed to find that psycho – social needs of the elderly have not been met and additional expense generated by elderly dependents have exacerbated their economic problems.
The introduction seems to suggest that the study is focused on the relationship between unmet needs of patients with chronic diseases in health care centers and their impact on health based on survey data, Or it is a study on how economic problems and loneliness pose limitations to adequately receive their subjective health needs as suggested by the title of the paper. However, this paper only presents the facts of whether health needs of chorionic patients are met and is devoid of discussion related to the consequences of their health. The author notes there are issues of economic problems due to dependence needs and psycho – social needs due to loneliness, but without further evidences or exploration on why this is a dominant problem to the health of choric diseases elderly.
Section 4 on the discussion seems to refer to background section. Most of the discussion and data in this section 4 does not relate on survey conducted in this study. It is difficult to understand how this section 4 is related to the main topic of this paper. In what ways do survey findings can support or improve our understanding on the issues in this section which have been well discussed in the other research.
There is a limited amount of sample data. Participants were selected from convenience, but the characteristics of the elderly in the sample (age, gender, and degree of dependence) are concentrated in distribution. There are obvious problems with the random selection of the samples, which directly leads to unconvincing findings.
I can understand that the issue on health needs is very important for chronic elderly patients, and I would expect that the finding and discussing are more focused on this group of people by emphasizing their special needs or characteristics. The current paper however to me is to generalize on all elderly needs without differentiation, except by their survey which is based on chorionic patients.
In short, the issues raised in the introduction are important and require comprehensive research. However, the data and methods used in the current paper are inconclusive, and the conclusions of the study lacks scientific theory and data support. The current paper leans toward a survey report than an academic article.
Author Response
Economic problems and loneliness as factors related to subjective unmet health needs in people with chronic diseases and dependence (IJERPH-754706)
Major revision 1 - Cover letter - Reviewer 1
Thank you very much for the suggested modifications for the paper Economic problems and loneliness as factors related to subjective unmet health needs in people with chronic diseases and dependence.
In the paper resubmitted you can find modifications done to other three reviewer´s suggestions (one different colour to each). In this case, we provide responses to your analysis, in order to clarify it. Please, let us know, more specifically, which modifications we should do to improve the paper.
Summary: this study aims to explore subjective unmet health needs through conducting a focus group survey for 23 elderly chorionic patients in day – care center. Qualitative exploratory – descriptive study is analyzed to find that psycho – social needs of the elderly have not been met and additional expense generated by elderly dependents have exacerbated their economic problems.
The introduction seems to suggest that the study is focused on the relationship between unmet needs of patients with chronic diseases in health care centers and their impact on health based on survey data, Or it is a study on how economic problems and loneliness pose limitations to adequately receive their subjective health needs as suggested by the title of the paper.
The aim of the study is to explore the perceptions of chronic patients over 65 years of age from the day-care center toward subjective health needs that are not being met by the socio-health system. The main results and the aspects that patients with chronic diseases most focalized were economic problems and loneliness. So, we have highlighted this main results in the title of the paper.
However, this paper only presents the facts of whether health needs of chorionic patients are met and is devoid of discussion related to the consequences of their health. The author notes there are issues of economic problems due to dependence needs and psycho – social needs due to loneliness, but without further evidences or exploration on why this is a dominant problem to the health of choric diseases elderly.
At the beginning of the study we didn´t know about this results. We are surprised about result obtained. In Discussion section we provide data and bibliography about it ¿Do you think it would be better to change the name of the study or to provide more bibliography?
Section 4 on the discussion seems to refer to background section. Most of the discussion and data in this section 4 does not relate on survey conducted in this study. It is difficult to understand how this section 4 is related to the main topic of this paper. In what ways do survey findings can support or improve our understanding on the issues in this section which have been well discussed in the other research.
The discussion section is focused on the main results: material of technical aid, psychological needs and psycho-social needs. And we have proportioned references in order to discuss these findings. Do you think we have to move some data or paragraphs to introduction section, or remove it?
There is a limited amount of sample data. Participants were selected from convenience, but the characteristics of the elderly in the sample (age, gender, and degree of dependence) are concentrated in distribution. There are obvious problems with the random selection of the samples, which directly leads to unconvincing findings.
The characteristics of the sample are the characteristics of the elderly people that met the inclusion criteria. One of the most important inclusion were the moderate or severe degree of dependence in terms of the Barthel Index. Moreover, all of them were day-care centre users. Both inclusion criteria caused the homogenization of the sample. It is noticed in limitations subheading also (line 400-401).
We are thinking about a similar research with a sample of elderly people with a mild degree of dependence and NON day-care centre users, in order to compare results.
I can understand that the issue on health needs is very important for chronic elderly patients, and I would expect that the finding and discussing are more focused on this group of people by emphasizing their special needs or characteristics. The current paper however to me is to generalize on all elderly needs without differentiation, except by their survey which is based on chorionic patients.
We have emphasized the findings and the discussion about them. It is clear that major differences doesn´t exist between elderly people´s health needs and people with chronic diseaase and dependence health needs. We agree with you, becasuse loneliness and economic problems, for example, generate the same problems in both groups, as we shows with references in discussion section.
In short, the issues raised in the introduction are important and require comprehensive research. However, the data and methods used in the current paper are inconclusive, and the conclusions of the study lacks scientific theory and data support. The current paper leans toward a survey report than an academic article.
Thank you for your suggestions.

Reviewer 2 Report
Thank you for the opportunity to review this important article. I have worked with this population for over 10 years and appreciate your work in this area. I found the manuscript to be well organized and written. Your methods are appropriate and complete. Thorough explanation of the results and discussion. Excellent manuscript.
My only suggestion is a few editing issues (very minor) to help with readability. There are a few very long sentences. For example, Page 8 lines 211-215. I had to read it several times to fully understand all the details. Another sentence is on Pages 9-10 lines 264-267. Again, it is a bit long with 'they' used 3 times and 'them' once. Three times you are referring the patients ('they' x2; 'them' x1) and once to the children of the patients. It took several readings to figure out who you were talking about in each pronoun.
'Author Contribution' section is from the journal template and does not include your author information.
Author Response
Economic problems and loneliness as factors related to subjective unmet health needs in people with chronic diseases and dependence (IJERPH-754706)
Major revision 1 - Cover letter - Reviewer 2
Thank you very much for the suggested modifications for the paper Economic problems and loneliness as factors related to subjective unmet health needs in people with chronic diseases and dependence. Having them already done, all the authors believe that the article has improved.
Every modification done is described below. In the paper you can find them in blue colour:
Thank you for the opportunity to review this important article. I have worked with this population for over 10 years and appreciate your work in this area. I found the manuscript to be well organized and written. Your methods are appropriate and complete. Thorough explanation of the results and discussion. Excellent manuscript.
My only suggestion is a few editing issues (very minor) to help with readability. There are a few very long sentences. For example, Page 8 lines 211-215. I had to read it several times to fully understand all the details.
Thank you very much for your support. We have modified this too long sentence in three shorter (lines 221-223).
Another sentence is on Pages 9-10 lines 264-267. Again, it is a bit long with 'they' used 3 times and 'them' once. Three times you are referring the patients ('they' x2; 'them' x1) and once to the children of the patients. It took several readings to figure out who you were talking about in each pronoun.
Rephrased in lines 272-275).
'Author Contribution' section is from the journal template and does not include your author information.
“Author Contribution” were provided in the submission proccess and it will be incorporated throughout the editorial process. Here you are the screenshot about it:
(You can see it in the pdf document sent).

Reviewer 3 Report
I felt that this was a good effort, and thus I feel inclined to suggest it is a good fit for inclusion within the journal. There are a few things worth noting however, which I have listed in order of appearance:
1) I think that you meant to say over 65 years of age in line 42.
2) I am not sure what the Barthel Index is (mentioned on p. 3). Please elaborate on this further.
3) I think you meant to say "Ethics Committee" on page 6, line 137.
4) I do not think that I have heard of the term "sociosanitary systems", and am wondering if it is largely a European term. This is probably a failing on my part.
5) The quote on page 8 from GF2PC-P5 related to children handling the finances did not necessarily sound like an unmet need from the quote provided. It almost sound like he/she welcomed the help.
6) On line 229, replace with "There were a series of critiques..."
7) I have issues with the statement on lines 306-307. This sounds like editorializing, rather than established fact.
8) You mentioned the Dependency Law pretty late in the manuscript. Please provide a description of that for others like me that do not know much about the law.
9) What do you mean by the statement on line 369? Whose definition of health are you referring to here?
10) You state twice in the document that participants did not have unmet basic physiological needs. In the next breath though, you state that the real issue was with technical assistance material. In my opinion, this still shows unmet basic physiological needs.
Author Response
Economic problems and loneliness as factors related to subjective unmet health needs in people with chronic diseases and dependence (IJERPH-754706)
Major revision 1 - Cover letter - Reviewer 3
Thank you very much for the suggested modifications for the paper Economic problems and loneliness as factors related to subjective unmet health needs in people with chronic diseases and dependence. Having them already done, all the authors believe that the article has improved.
Every modification done is described below. In the paper you can find them in green colour:
I felt that this was a good effort, and thus I feel inclined to suggest it is a good fit for inclusion within the journal. There are a few things worth noting however, which I have listed in order of appearance:
- I think that you meant to say over 65 years of age in line 42.
Sorry, is over 15 years. The last European Health Interview Survey took into account people over 15 years. Here you are an excerpt of the methodologie of survey about the age of population interviewed.
(In the pdf document you can see a screenshot about it).
- I am not sure what the Barthel Index is (mentioned on p. 3). Please elaborate on this further.
We have added a brief explanation about it (lines 105-107).
- I think you meant to say "Ethics Committee" on page 6, line 137.
Yes. Sorry. Modified (line 147).
- I do not think that I have heard of the term "sociosanitary systems", and am wondering if it is largely a European term. This is probably a failing on my part.
Sorry. “Socio-sanitary system” is a synonym term for “socio-health support”, “socio-sanitary service” or “health and social support system”.
- The quote on page 8 from GF2PC-P5 related to children handling the finances did not necessarily sound like an unmet need from the quote provided. It almost sound like he/she welcomed the help.
With this quote we have highlighted how a person has lost the control of their economy, as the name of the subtheme says (Loss of control on their own economy) ¿It would be neccesary to replace it?
- On line 229, replace with "There were a series of critiques..."
Thank you for locating the mistake. Replaced (line 237).
- I have issues with the statement on lines 306-307. This sounds like editorializing, rather than established fact.
Sorry, but we don´t know the exact statement and what should we do.
- You mentioned the Dependency Law pretty late in the manuscript. Please provide a description of that for others like me that do not know much about the law.
Thank you for your observation. We have added a brief explanation about this law (lines 361-365).
- What do you mean by the statement on line 369? Whose definition of health are you referring to here?
We have highlighted is The World Health Organization´s definition of health (line 382).
- You state twice in the document that participants did not have unmet basic physiological needs. In the next breath though, you state that the real issue was with technical assistance material. In my opinion, this still shows unmet basic physiological needs.
Yes. We agree. And in this way, we state it in the first paragraph under subheading 3.1. Biological needs results. So, we have added a sentence in Discussion section in order to clarify it (line 305).

Reviewer 4 Report
Please add information to make it easy to understand you result.
1) Information about PCDD and day centers
We do not know "who is the users of day centers," "what is day centers in your country," and "how many PCDD usually uses day centers." This information can show the goodness of your sampling measure.
2) When Pilot-FGI and FGIs were done
3) Please explain the researchers' background as readers can assess how your background might affect your analysis. I think the information about the researcher's experience around the day center will be helpful.
3) I do not think all dendrograms (figure 1-5) are more reasonable than tables. I recommend you to remove them and show all categories by using tables. It can help readers to understand your "comprehensive" result.
4) "Author contributions" should be changed along with your project.
5) The FGI tools 2.5-3hours, I think it might be too long for PCDD. Please add how you considered their safety.
5) Please modify the introduction part
The reason why your research question focused on "perception of unmet needs" is not clear. Your result can be read as simply classified participants' expression referring to "the already know (existing) types of needs." I am afraid that the results can not meet your interest as RQ."
6) Please clarify what your research newly found, in the discussion part
Author Response
Economic problems and loneliness as factors related to subjective unmet health needs in people with chronic diseases and dependence (IJERPH-754706)
Major revision 1 - Cover letter - Reviewer 4
Thank you very much for the suggested modifications for the paper Economic problems and loneliness as factors related to subjective unmet health needs in people with chronic diseases and dependence. Having them already done, all the authors believe that the article has improved.
Every modification done is described below. In the paper you can find them in red colour:
Please add information to make it easy to understand you result.
1) Information about PCDD and day centers
We do not know "who is the users of day centers," "what is day centers in your country," and "how many PCDD usually uses day centers." This information can show the goodness of your sampling measure.
We have added this information in Introduction section (lines 59-62).
2) When Pilot-FGI and FGIs were done
Detailed in lines 118 and 121.
3) I do not think all dendrograms (figure 1-5) are more reasonable than tables. I recommend you to remove them and show all categories by using tables. It can help readers to understand your "comprehensive" result.
Removed figures 1-5. Tables 3-7 created.
4) "Author contributions" should be changed along with your project.
Yes. All authors have done writing-original and writing-review.
5) The FGI tools 2.5-3hours, I think it might be too long for PCDD. Please add how you considered their safety.
This duration of the focus groups (1.5-2.5 hours) includes some breaks. It is the total duration of the recordings (line 128).
5) Please modify the introduction part
The reason why your research question focused on "perception of unmet needs" is not clear. Your result can be read as simply classified participants' expression referring to "the already know (existing) types of needs." I am afraid that the results can not meet your interest as RQ."
We explain the importance of subjetive unmet needs in lines 63-69, and references about are provided in lines 70-76. So, we have added a paragraph focused on it importance (lines 77-81).
6) Please clarify what your research newly found, in the discussion part
Thank you for your suggestion. We have added a paragraph in order to clarify it at the end of Discussion section (lines 387-393).

Round 2
Reviewer 1 Report
I cannot see any changes based on my comments, even the author agrees with most of my opinions. In this case, I still think that the objective of the paper cannot be supported by the data. The method in this article cannot achieve convincing results. To me, the current paper still looks like a survey report, not an academic research article.
Author Response
Economic problems and loneliness as factors related to subjective unmet health needs in people with chronic diseases and dependence (IJERPH-754706)
Major revision 2 - Cover letter - Reviewer 1
Thank you very much for the suggested modifications for the paper Economic problems and loneliness as factors related to subjective unmet health needs in people with chronic diseases and dependence.
I cannot see any changes based on my comments, even the author agrees with most of my opinions. In this case, I still think that the objective of the paper cannot be supported by the data. The method in this article cannot achieve convincing results. To me, the current paper still looks like a survey report, not an academic research article.
We have added the Author Contributions.
In revision 1, you provided some comments about this paper. But your comments you provided very general comments without specifying which modifications we could make to improve it. In our responses, some paragraphs of the paper were indicated in relation to your comments. But this research was carried out using a qualitative methodology and to give concrete answer it is difficult.
On this basis, we hope that this obstacle can soon be cleared.
